# Identification of Marker Genes in Infectious Diseases from ScRNA-seq Data Using Interpretable Machine Learning

**DOI:** 10.3390/ijms25115920

**Published:** 2024-05-29

**Authors:** Gustavo Sganzerla Martinez, Alexis Garduno, Ali Toloue Ostadgavahi, Benjamin Hewins, Mansi Dutt, Anuj Kumar, Ignacio Martin-Loeches, David J. Kelvin

**Affiliations:** 1Microbiology and Immunology, Dalhousie University, Halifax, NS B3H 4H7, Canada; gustavo.sganzerla@dal.ca (G.S.M.); ali.toloue@dal.ca (A.T.O.); benjamin.hewins@dal.ca (B.H.); mansidutt@dal.ca (M.D.); kumaranuj@dal.ca (A.K.); 2Department of Pediatrics, Izaak Walton Killam (IWK) Health Center, Canadian Center for Vaccinology, Halifax, NS B3H 4H7, Canada; 3Department of Immunology, Shantou University Medical College, Shantou 512025, China; 4Department of Clinical Medicine, Trinity College Dublin, D08 NHY1 Dublin, Ireland; gardunoa@tcd.ie (A.G.); imartinl@tcd.ie (I.M.-L.); 5Department of Intensive Care Medicine, St. James’s Hospital, D08 NHY1 Dublin, Ireland; 6Multidisciplinary Intensive Care Research Organization (MICRO), St. James’s Hospital, D08 NHY1 Dublin, Ireland

**Keywords:** sepsis, single-cell RNA sequencing, marker genes, artificial intelligence

## Abstract

A common result of infection is an abnormal immune response, which may be detrimental to the host. To control the infection, the immune system might undergo regulation, therefore producing an excess of either pro-inflammatory or anti-inflammatory pathways that can lead to widespread inflammation, tissue damage, and organ failure. A dysregulated immune response can manifest as changes in differentiated immune cell populations and concentrations of circulating biomarkers. To propose an early diagnostic system that enables differentiation and identifies the severity of immune-dysregulated syndromes, we built an artificial intelligence tool that uses input data from single-cell RNA sequencing. In our results, single-cell transcriptomics successfully distinguished between mild and severe sepsis and COVID-19 infections. Moreover, by interpreting the decision patterns of our classification system, we identified that different immune cells upregulating or downregulating the expression of the genes *CD3*, *CD14*, *CD16*, *FOSB*, *S100A12*, and *TCRɣδ* can accurately differentiate between different degrees of infection. Our research has identified genes of significance that effectively distinguish between infections, offering promising prospects as diagnostic markers and providing potential targets for therapeutic intervention.

## 1. Introduction

Infectious diseases remain a substantial challenge for global health, a reality that was underscored by the emergence of the most devastating pandemic in over a century. This pandemic was caused by the infectious agent SARS-CoV-2, which brought viruses into focus as a significant threat to public health worldwide. For severe infections, a “cytokine storm” emerges when the immune system’s response to an infectious entity becomes so intense that it inadvertently turns against the well-being of the host. Moreover, cytokine storms contribute to multi-organ failure. Sepsis arises from the extreme response of the body to an infectious agent—be it a virus, bacterium, or fungus—manifesting as extensive inflammation that may cause organ dysfunction [1,2]. It is estimated that up to a third of intensive care unit (ICU) patients contend with sepsis [3]. While fatality rates fluctuate across nations and regions, the year 2017 saw approximately 11 million sepsis-related deaths amidst 48.9 million newly diagnosed cases globally, translating to a mortality rate of 22% [4]. Moreover, projections posit that sepsis accounted for approximately 18 to 21% of the total global deaths during the same period.

The dysregulation of the immune system while trying to tackle an infectious agent leaves behind footprints that can be used to identify markers of severity. In analyzing the concentration of proteins in the bloodstream of infected individuals, markers of vascular transformation, inflammation, oxidative stress, and chemotaxis have successfully been employed as good descriptors of a dysregulated immune response [5,6].

Here, we highlight the single-cell RNA sequencing (scRNA-seq) technique, which allows for the analysis of gene expression at a cellular level. Many areas of research and clinical practice can benefit from scRNA-seq, such as cancer research, drug discovery and development, and immunology, among others. One remarkable translational advantage of scRNA-seq is the development of precise therapeutic strategies [7]. In order to benefit from granularity of data from scRNAseq, exquisite data analytics techniques are needed [8]. The analysis of genomics data might be powered by artificial intelligence (AI) techniques, which hold the mathematical power to untangle the intricate relationship between data points in higher-dimensional settings [9]. Examples of AI-assisted medicine can be found in the identification of marker genes [10,11], biomarkers [12,13], vaccine candidates [14,15], and potential targets for therapeutics [16]. Moreover, AI has been used in conjunction with scRNA-seq data analysis for the identification of tumor cells [17]. AI generative models have also been employed in predicting the expression level of genes in a single cell [18].

In our work, we hypothesize that AI has the capacity to untangle patterns and capture signals of individual cells expressing different transcripts when stressed by moderate or severe manifestations of viral- or bacterial-induced sepsis. Moreover, by mapping the decision patterns of AI tools, we can generate valuable insights into the intricate immunological responses deployed by the body in its battle against infectious pathogens, highlighting essential marker genes for advancing therapeutic interventions. 

## 2. Results

### 2.1. Clinical Characteristics of the Cohort

We acquired data from 16 patients, 4 of whom had sepsis, 4 had moderate COVID-19, 4 had septic shock, and 4 had severe COVID-19 (Table 1). We found the average age of the patients to be 65 years old. The male-to-female ratio in this cohort was roughly two to one. Recordings showed temperatures higher than 37 °C with the septic shock population showing the highest mean (39.7 °C). Patients with mild COVID-19 had the highest P/F ratio, while those with sepsis and septic shock had the lowest mean. Neutrophil numbers were found to be the lowest in moderate COVID-19 patients and the greatest in sepsis patients. The sepsis group had the lowest mean lymphocyte count. The lowest Sequential Organ Failure Assessment (SOFA) score was observed in individuals with mild COVID-19 and the highest in those with septic shock. It is worth noting that severe COVID-19 patients had a SOFA score that was even lower than those suffering from sepsis.

### 2.2. Using scRNA-seq Transcripts as an Input to Classify Diagnostics

We created a function of 20 transcripts to classify the diagnostics of patients (i.e., mild sepsis, mild COVID-19, septic shock, and severe COVID-19). Upon testing the classification capacity of Random Forest (RF), Support Vector Machines (SVM), Extreme Gradient Boosting (XGBoost), Linear Regression (LR), Gradient Boosting (GB), and K-Nearest_Neighbors (KNN), we determined that the best classification metrics were achieved by XGBoost (Table 2), in which we show the performance metrics of different machine learning models in classifying scRNA-seq data from 37,732 cells from 16 different patients. As we have a multiclass classification problem (sepsis, mild COVID-19, septic shock, and severe COVID-19), we used a one-versus-all classification. 

Once XGBoost was selected as the classification algorithm to convey this analysis, we identified its best parameters, i.e., number of estimators = 200, learning rate = 0.2, and maximum depth = 7. The input transcripts successfully classified the mild sepsis, mild COVID-19, septic shock, and severe COVID-19 patients in a one-versus-all approach by obtaining an AUC score of 0.93, 0.91, 0.89, and 0.96, respectively (Figure 1A). Moreover, we mapped the decision pattern of our XGBoost classifier with SHAP to identify the global and local contribution of each transcript in predicting a class (Figure 1B–F). Here, we list the transcripts *CD16*, *TCRɣδ, CD14, FOSB, CD3*, and *S100A12* as being among the top six most important features across the four medical conditions. Also, their expression levels (up or downregulated) could distinguish between diagnostics. First, we identified that the higher expression of *CD3* is a characteristic of the severe forms of sepsis and COVID-19; this same pattern was observed for the gene *CD14*. Next, we found the lowest levels of *CD16* in cells from sepsis patients, while the highest levels were found in mild COVID-19 patients. The gene *FOSB* was found to be expressed in higher amounts in the mild forms of sepsis and COVID-19. Finally, a similar pattern was found for the expression of the genes *S100A12* and *TCRɣδ*, in which they were highly expressed in the bacterial forms of infection and lowly expressed in the viral infections.

We performed a differential expression analysis to check the level of each transcript according to the cell type it came from across sepsis, mild COVID-19, septic shock, and severe COVID-19 patients. We limited the analysis to the six genes that were listed as important descriptors of each medical condition, i.e., *CD3*, *S100A12*, *FOSB*, *CD14*, *TCRɣδ*, and *CD16*. The results show important signals that differentiate sepsis (Figure 2A), mild COVID-19 (Figure 2B), septic shock (Figure 2C), and severe COVID-19 (Figure 2D). First, the *CD16* gene is shown to express higher in sepsis and mild COVID-19 patients, particularly across dendritic and *ɣδ* T cells. The same profile of *CD16* expression was found to be lower in the same cell types across severe manifestations of septic shock and severe COVID-19, suggesting *CD16* as a marker gene for severity. Moreover, the transcript *CD3* was shown to represent a higher expression normalized value in septic shock, particularly when subgrouping *ɣδ* T cells. This transcript was also upregulated by *CD4* memory T cells, dendritic cells, and *ɣδ* T cells in severe COVID-19 patients, while *CD3* showed a lower expression rate across all cell types in sepsis and mild COVID-19 patients. *S100A12* has shown very low expression in mild COVID-19 patients and no expression in severe COVID-19 patients, particularly when subgrouping differences across CD8 memory T cells and *ɣδ* T cells, meaning it could be a representative target in viral-induced diseases. Interestingly, in the case of septic shock patients, *S100A12* served as a relevant gene target when analyzing monocyte non-classical cells. Another result worth mentioning is the clear upregulation of *TCRɣδ* in severe COVID-19 patients, particularly when looking at differences in dendritic cells compared to mild COVID-19 and bacterial-induced sepsis, making it a potential classifier in critically ill COVID-19 patients.

### 2.3. Using scRNA-seq Transcripts as an Input to Classify Cell Types

In order to assess whether gene transcripts are a good indicator for different cell types, we performed an XGBoost multiclass classification where the inputs are scRNA-seq transcripts and the output is B, dendritic, monocyte classical, monocyte nonclassical, natural killer, TCD4 memory and naïve, TCD8 memory and naïve, and *ɣδ* T cells. Our model (Figure 3A) could distinguish between all cell types with AUC scores ranging from 0.96 (highest, *ɣδ* T cells) to 0.75 (lowest, TCD8 memory). Moreover, we checked the individual contribution of transcripts to assign a label to a cell type in the classification models with an AUC higher than 0.8. When looking at the gene transcripts that best characterized each cell type, we identified the transcripts *DEFA3* and *FOSB* to be the best indicators for dendritic cells (Figure 3C), monocyte classical cells (Figure 3D), monocyte nonclassical cells (Figure 3E), TCD4 memory cells (Figure 3G), TCD8 naïve cells (Figure 3H), and *ɣδ* T cells (Figure 3I). We also identified that the expression of *CD24* is a good indicator for B cells (Figure 3B). 

### 2.4. GO Analysis of Shortlisted Candidate Genes

Based on the expression profiling of genes in various cell types under diverse medical conditions, six genes were shortlisted for the GO analysis. The GO analysis predicted enriched biological processes (BPs), molecular functions (MFs), and cellular components (CCs) for five of these genes (*CD3*, *S100A12*, *FOSB*, *CD14*, and *CD16*). However, during the GO analysis, the *TCRɣδ* gene did not map to the database, so we excluded it from the analysis. The GO analysis revealed that these candidate genes are associated with a wide range of BPs, including responses to external stimuli, immune system processes, immune responses, immune effector processes, responses to stress, regulation of responses to immune system processes and stimuli, and regulation of signaling, among others (Figure 4a). The most highly enriched MFs associated with these candidate genes included identical molecular transducer activity, signaling receptor activity, protein-containing complex binding, structural molecular activity, DNA-binding transcription factor activity, and lipid binding, among others (Figure 4b). Furthermore, the GO analysis identified the extracellular region, extracellular space, cell surface, side of the membrane, and organelle membrane as the most enriched CCs for these candidate genes (Figure 4c). As evident from Figure 4d, the Kyoto Encyclopedia of Genes and Genomes (KEGG) pathway analysis indicated that these five candidate genes are involved in various metabolic pathways, including hematopoietic cell lineage, osteoclast differentiation, phagosome formation, antigen processing and presentation, etc. 

### 2.5. Summary of Cellular Expression Patterns for Various Diagnostics in a Two-Dimensional Format

To analyze 37,732 cells from 16 patients with different medical conditions expressing 20 different immunological transcripts, a UMAP dimensionality reduction technique was applied. First, we determined whether the transcripts could provide a visualization of the patients according to their diagnosis (Figure 5A). The major differences were that cells from sepsis patients on the left-hand side of the image were well separated from other cells. Also, in the middle of the plot, cells from all four conditions were exhibiting similar behavior. Furthermore, on the right-hand side of the image, we noticed cells from sepsis and severe COVID-19 patients behaving similarly. Second, we investigated the cell type in question. We found a predominance in monocyte classical cells as well as CD8+ naïve T cells (Figure 5B). Third, we checked whether the expression profile of the cells would be able to visually distinguish between severe and non-severe patients (Figure 5C). We found islands on the left-hand side of the image, where there were only cells from non-severe patients. Lastly, we investigated whether the transcription by cells from COVID-19 patients would be distinct from non-COVID-19 patients (Figure 5D). We found the same profile on the left-hand side of the image, where there were only cells from non-COVID-19 patients expressing similar behavior. 

## 3. Discussion

In our investigation, we have identified significant transcriptomic alterations, particularly in the genes *CD3*, *CD14*, *CD16*, *FOSB*, *S100A12*, and *TCRɣδ*, that differentiate sepsis, mild COVID-19, septic shock, and severe COVID-19. Here, we highlight the importance of interpreting these transcript signals, as they may lead to the identification of potential marker genes associated with severity. The heightened expression of the *CD3* gene has been identified and delineated as a significant indicator of severe COVID-19. This gene is responsible for encoding a cluster of proteins that collectively form the T cell receptors, which are crucial for activating T cells and promoting their cytotoxicity. Notably, upon inhibiting the expression of *CD3*, a controlled inflammatory response was observed in the context of COVID-19 [19]. Thus, we theorize that the upregulation of *CD3* that differentiated severe COVID-19 might have caused the cytotoxic effect of CD3+ T cells attacking infected host cells. This becomes more apparent as none of the septic shock patients were reported as being infected with intracellular bacteria, which trigger the immune system to attack the infected host cells.

Our study reveals consistent upregulation of the *CD14* gene in the severe manifestations of both viral and bacterial infections. This gene encodes a cell surface receptor that differentiates monocytes into macrophages with the ability to recognize pathogens and trigger immune responses. In a bacterial infection, CD14^+^ cells will recognize the lipopolysaccharides (LPS) found in the bacterial cell wall of gram-negative bacteria to trigger an immune response. In a viral infection, CD14+ cells also express other pattern recognition receptors that identify specific viral signals, such as their unique genetic material configuration. These signals are part of the pathogen-associated molecular patterns (PAMPs), or they may arise from infected cells producing signals categorized as damage-associated molecular patterns (DAMPs) [20]. Here, we suggest that the increase of *CD14* could potentially indicate severity in certain conditions. Firstly, in septic shock, it is believed that CD14+ cells binding with LPS play a role, particularly as septic shock patients often show higher microbial levels. Secondly, in severe cases of COVID-19, it is proposed that the higher levels of viral RNA in the bloodstream necessitate more CD14-positive cells to identify these viral molecular patterns, as indicated in a study by [21]. In both instances of infection, we identified the differentiated upregulation of *CD14* as a marker of severity, supported by previous evidence [22] on the inhibition of the upregulation of *CD14* as a potential therapeutic against severe infection.

We observed the downregulation of *CD16* as a marker of severe COVID-19 (compared to mild COVID-19). The protein encoded by this gene is an antibody Fc receptor. The pathogenesis of COVID-19 has been connected to transcriptional alterations in *CD16* [23,24]. By lowering its expression, this gene prevents macrophages and NK cells from producing the FcɣRIIIA receptor, which decreases the binding of IgG antibodies to immune cells. This situation might contribute to a more severe infection, as shown in patients with severe COVID-19. We also found lower expression of *FOSB* in severe COVID-19 and septic shock patients. *FOSB* encodes the protein AP-1, a class of transcription factors (TFs). Although the AP-1 TF is known to stimulate the expression of downstream genes, there is evidence describing this TF as being a repressor as well [25]. According to our data, *FOSB* expression is inversely associated with prognosis; thus, it might be the case that lower *FOSB* expression is preventing potential pathways that lead to a dysregulated immune response being repressed.

Our model highlights the substantial upregulation of the *S100A12* gene, which encodes the small protein Calgranulin C in patients with sepsis and is more pronounced in patients with septic shock. Calgranulin C, primarily expressed by neutrophils, can elicit cytokine production upon extracellular secretion, thereby contributing to pro-inflammatory activities [26]. Interestingly, we found that both sepsis and septic shock patients with higher expression of *S100A12* also have a higher neutrophil count, a cell type whose immune role is well documented due to its proficiency in bacterial clearance through phagocytic mechanisms [27]. These findings underscore the significance of upregulated *S100A12* as a potential marker of bacterial infections. We identified multifaceted expression levels regarding *TCRɣδ*, a membrane receptor of T cells. In bacterial infections, we found the expression of *TCRɣδ* is negatively correlated with severity. A study found deceased ICU patients presenting T lymphocytopenia also had a steep decline in the expression levels of ɣδ+ T cells [28,29]. In viral infections, we found the opposite, where the expression of *TCRɣδ* is positively correlated with severity. The association of ɣδ+ T cells and acute COVID-19 has been explored [30], despite it being unclear how the level of *TCRɣδ* at the start of the disease influences the outcome.

Despite analyzing transcriptional changes in over 37,000 cells from patients with varied conditions, our study faced limitations due to the small sample size of our cohort. We believe that by having more patients, our machine learning model can undergo different layers of validation, granting even more robust results. Also, we were limited to investigating a panel containing 400 transcripts associated with immune functions. Even though this would be out of the scope of this study, we believe that the complexity of immune responses could be reflected in transcriptional changes in genes beyond the initial scope of our ScRNA-seq analysis, providing a more comprehensive perspective on the expression levels of the analyzed patients. Moreover, our study aimed to analyze the differential expression of genes among different subtypes of infection, and we posit that to elucidate the mechanistic role of upregulation and downregulation of our identified genes, an in-depth review of their role in the context of infection is needed.

In conclusion, our investigation utilized ScRNA-seq in conjunction with explicable artificial intelligence. The synergy between these methodologies has the potential to shed light on the processes that are responsible for immune responses by identifying transcriptional alterations in genes of interest during moderate and severe presentations of infectious diseases.

## 4. Materials and Methods

### 4.1. Patient Enrollment and ScRNA-seq

Peripheral blood mononucleated cells were profiled from 16 patients hospitalized in the St. James’s Hospital in Dublin (Republic of Ireland). The recruitment included eight septic critically ill patients with multiorgan failure (four from SARS-CoV-2 and four from bacterial sepsis) and eight septic patients with single organ failure (four from SARS-CoV-2 and four from bacterial sepsis). Blood samples from the 16 patients were collected into 10 mL EDTA tubes (Becton, Dickinson and Co., Franklin Lakes, NJ, USA). These samples were then loaded into SepMate 50-mL tubes (Stemcell Technologies, France) for further processing. The tubes were centrifuged at 1200× *g* for 10 min at room temperature with the brake engaged. After centrifugation, the cell interface was extracted using a pipette and subsequently washed using PBS-EDTA through centrifugation at 400× *g* for 7 min. The resulting PBMC pellets were suspended in an ammonium chloride solution (Stemcell Technologies, France) and allowed to incubate for 10 min at room temperature on a mixing platform. This step aimed to lyse the remaining red blood cells. Following the lysis process, the lysed pellets were resuspended in 10 mL of PBS, with a small sample set aside for cell counting purposes. The resuspended mixture underwent another round of centrifugation at 400× *g* for 7 min, still with the brake applied. The isolated PBMCs were then subjected to another wash with PBS-EDTA before being resuspended for downstream analyses. All these processing steps were carried out within 4 h of collecting the blood samples. Moreover, samples obtained on the same day were processed simultaneously to mitigate any variations stemming from the processing timeline. To ensure optimal viability of cells and prevent cell clumping, the samples were analyzed in their fresh state using the BD Rhapsody Single-Cell Analysis System platform. One BD Rhapsody cartridge was loaded with 10,000 polled cells from each patient. The single cells were isolated through single-cell capturing and cDNA synthesis with the BD Rhapsody Express Single-Cell Analysis System following the manufacturer’s recommendations (BD Biosciences, Franklin Lakes, NJ, USA). An immunological panel containing 400 transcripts associated with immune function targeted the sequenced transcripts through the amplification kit (cat. 633774, BD Biosciences, Franklin Lakes, NJ, USA). Quantity and quality control of the DNA were performed using the QubitTM dsDNA HS Assay Kit (cat. # Q32851, ThermoFisher Scientific, Waltham, MA, USA) and the electrophoresis system Agilent 2200 TapeStation cartridge (cat. #5067–5584, Agilent, Santa Clara, CA, USA). 

### 4.2. ScRNA-seq Data Analysis

We obtained single-cell expression matrices from 16 patients. Each row of the matrix represents a single cell. The matrices are composed of 400 columns representing a panel with 400 immune-related transcripts, from which we extracted the transcripts that are shared between all patients (variables *x*). From a classification point of view, each cell needs to have a variable *y* representing its label, which is inherent to the data. We employed two distinct classification routines, using the complex transcriptional signal represented by the transcripts *x* to classify: (i) a *y* cell type of B cell, dendritic, monocyte classical, monocyte nonclassical, natural killer, TCD4 memory, TCD8 naïve, or *ɣδ* T cells; and (ii) a *y* medical condition the cells are undergoing, i.e., sepsis, mild COVID-19, septic shock, and severe COVID-19. Both labels (cell type and medical condition) are already present in the data, where the cell types are obtained from the BD Rhapsody Single-Cell Analysis System platform and the medical condition matches that of the patient source.

The matrices were treated with in-house Python (version 3.9) scripts for merging the matrices, finding and sub-setting the matrices as per the transcripts in common, mapping the cell index into 10 different cell types, adding a column related to the parameters, and normalizing the data. A two-dimensional representation of 37,732 cells expressing 20 different immune-related genes was achieved by applying the Uniform Manifold Approximation and Projection (UMAP) algorithm implemented in combination with the Python packages Scanpy (version 1.9.3) and UMAP (version 0.0.1). The UMAP parameters followed: 40 neighbors, 0.2 of minimum distance, 2 dimensions in the embedding, and the distance metric between the datapoints chosen was Euclidean. The code that performs the ScRNA-seq analysis is publicly available at https://github.com/gustavsganzerla/immuno-scRNAseq/blob/main/single_cell.py, accessed on 24 May 2024.

The scanpy object was submitted using BBrowserX (version V.25). Filtering settings were also employed to ensure that only high-quality cells were retained. The settings include a minimum and maximum number of reads (10–1,000,000), a minimum and maximum number of identified genes (10–100,000), and a maximum percentage of mitochondrial genes of 25%. The normalized data were scaled further so that the mean expression across cells was 0 and the variance was 1.

### 4.3. Machine Learning Classification 

To classify 37,732 cells from 16 different patients with distinct diagnoses, we performed a multiclass classification assay. We divided the data into a proportion of 80:20 for training and testing, respectively; as we have four classes, we measured the classification performance of each class in a one-versus-all approach. For determining the best classifier, we tested in a Python environment the algorithms SVM with the radial basis function (RBF) kernel (scikit-learn version 1.3.0), RF (scikit-learn version 1.3.0), XGBoost (xgboost version 1.7.6), LR (scikit-learn version 1.3.0), GB (scikit-learn version 1.3.0), and KNN (scikit-learn version 1.3.0). Each algorithm was assessed for accuracy, precision, recall, and specificity. The algorithm that presented the best performance was chosen, and its hyperparameters were selected using the GridSearchCV method (scikit-learn version 1.3.0). Also, the model with the overall best performance received its false positive (FP) to false negative (FN) ratio through the analysis of the area under the curve (AUC) when classifying each class individually. Moreover, we used Shapley Additive Explanations (SHAP version 0.45.1) [31] to assess how the classifier made its decisions in classifying each class. The explanation process was applied to the training data. We assessed the extent to which the features (i.e., 20 immunological transcripts) were used for assigning a label to each class. We used a tree explainer (shap.TreeExplainer version 0.45.1), which quantifies the contribution of each feature to the model output on a per-instance basis. The idea behind the SHAP explanations is that the SHAP value of a feature is the average difference in prediction when that feature is included compared to when it is excluded, considering all combinations of features. The calculation of a SHAP value for a specific feature of a specific instance follows Equation (1).
$${SHAP}_{i}=\Sigma_{S\subseteq N\{i\}}\frac{\left|S\right|!\left(\left|N\right|-\left|S\right|-1\right)!}{|N|!}\left[f\left(x_{S}\cup \left\{i\right\}\right)-f(x_{S})\right]$$
where $f\left(x_{S}\cup \left\{i\right\}\right)$ is the model prediction for when feature i is included in the set of features S.

$f\left(x_{S}\right)$ is the prediction of the model when feature i is excluded from the set of features S.

N is the set of all features.

S is a subset of all features, excluding feature i.

$\left|S\right|$ is the number of features in subset S.

$\left|N\right|$ is the total number of features.

The code that implements the classification routine is available at https://github.com/gustavsganzerla/immuno-scRNAseq/blob/main/xgb_cell_classifier_w_shap.py, accessed on 24 May 2024.

### 4.4. Gene Ontology Analysis 

Gene ontology (GO) enrichment analysis of the shortlisted six genes, namely, *CD3*, *CD14*, *CED16*, *FOSB*, *S100A12*, and *TCRɣδ*, was performed with the aid of ShinyGO (version 0.77, http://bioinformatics.sdstate.edu/go/ accessed on 21 September 2023). During analysis, the false discovery rate (FDR) cutoff was set at 0.05 for the prediction of biological processes (BPs), molecular functions (MFs), metabolic pathways, and cellular components (CCs) of the considered genes. 

## Figures and Tables

**Figure 1 ijms-25-05920-f001:**
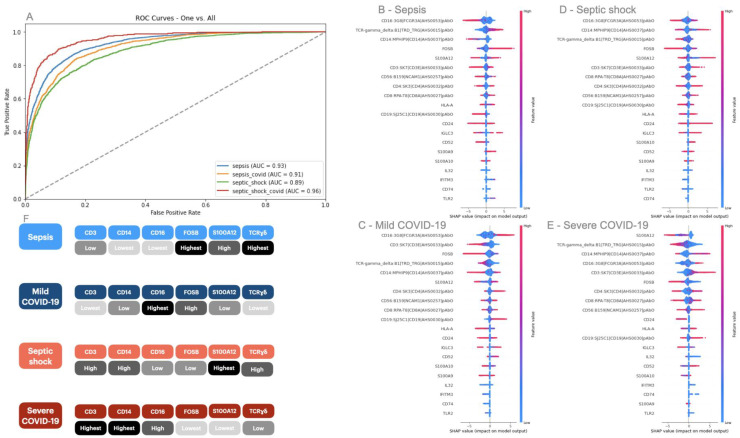
XGBoost classification and SHAP analysis of genes responsible for characterizing sepsis, septic shock, and moderate and severe COVID-19. In (**A**), we show the Area Under the Curve (AUC) score of an XGBoost multiclass classifier whose input is the expression level of immunological cells expressing 20 genes (i.e., *FOSB*, *IGLC3*, *S100A9*, *S100A10*, *CD56*, *IFITM3*, *CD52*, *IL32*, *CD74*, *S100A12*, *HLA-A*, *CD24*, *CD8*, *CD14*, *CD19*, *TCRɣδ*, *CD3*, *TLR2*, *CD16*, and *CD4*) mapped to a disease model (i.e., sepsis, septic shock, moderate COVID-19, and severe COVID-19). In (**B**–**E**), we show the SHAP explanation for each input gene in classifying sepsis, mild COVID-19, septic shock, and severe COVID-19. The results displayed in the SHAP plots consist of an input gene (**left** *y*-axis) as per the feature importance for a particular class, the expression of the gene (**right**
*y*-axis), and how this level of expression contributed to assigning a SHAP value (*x*-axis). Expression levels whose SHAP value is positive are significant descriptors of a class (i.e., disease model). Finally, in (**F**), we quantified the raw expression value of a subset containing the first five genes that best described each class.

**Figure 2 ijms-25-05920-f002:**
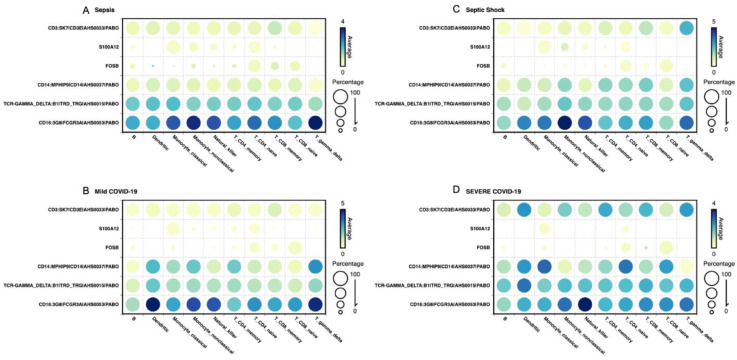
The study compared hub gene expression in mild (**A**,**B**) and severe cases (**C**,**D**) of viral- and bacterial-induced sepsis, respectively. We processed the data using a Seurat object and analyzed it with BBrowserX (version V.25) and created the visualization in BioVinci from BioTuring. https://bioturing.com. We found that certain genes, like CD16:3G8|FCGR3A|AHS0053|PABO, were more expressed in mild COVID-19 patients and sepsis, while CD3:SK7|CD3E|AHS0033|PABO showed higher levels in septic shock and severe COVID-19 cases. S100A12 had low expression in mild COVID-19 cases and none in severe cases, making it a potential target in viral-induced diseases. TCR-GAMMA_DELTA:B1|TRD_TRG|AHS0015|PABO was upregulated in severe COVID-19 cases, particularly in dendritic cells, suggesting its role as a classifier in critically ill COVID-19 patients.

**Figure 3 ijms-25-05920-f003:**
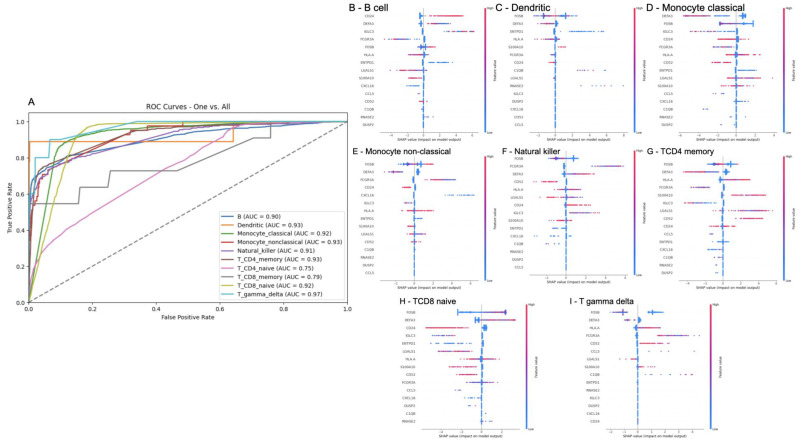
XGBoost classification of cell types and the interpretation of the classifier. In (**A**), we show the AUC plots of an XGBoost multiclass classifier using the transcripts of single cells to classify their corresponding type in a one-versus-all approach. The classes with an AUC equal to or higher than 0.8 were selected to have their model interpreted. In (**B**–**I**), we show the contribution of input features (i.e., transcripts) was accounted for by the assignment of B cell, dendritic, monocyte classical, monocyte nonclassical, natural killer, TCD4 memory, TCD8 naïve, and T gamma delta, respectively.

**Figure 4 ijms-25-05920-f004:**
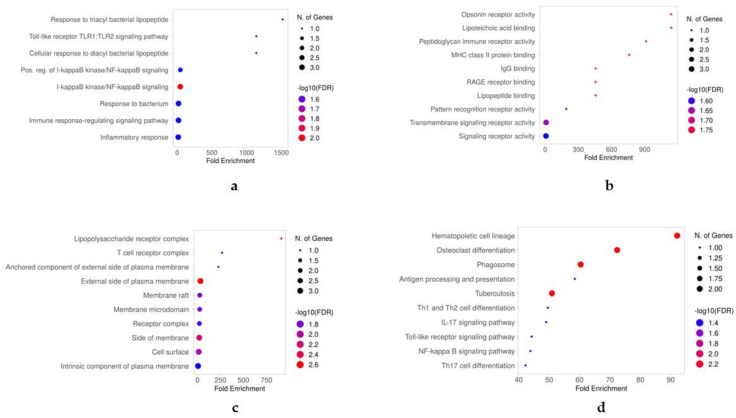
Gene ontology analysis of the genes *CD3*, *S100A12*, *FOSB*, *CD14*, and *CD16*. In (**a**), we show the biological processes mapped to the candidate genes. In (**b**), we show the molecular functions. The cellular components and enriched KEGG pathways are shown in (**c**,**d**), respectively. The results of the GO analysis are presented in the dot plots, and the GO terms were shortlisted based on the FDR values.

**Figure 5 ijms-25-05920-f005:**
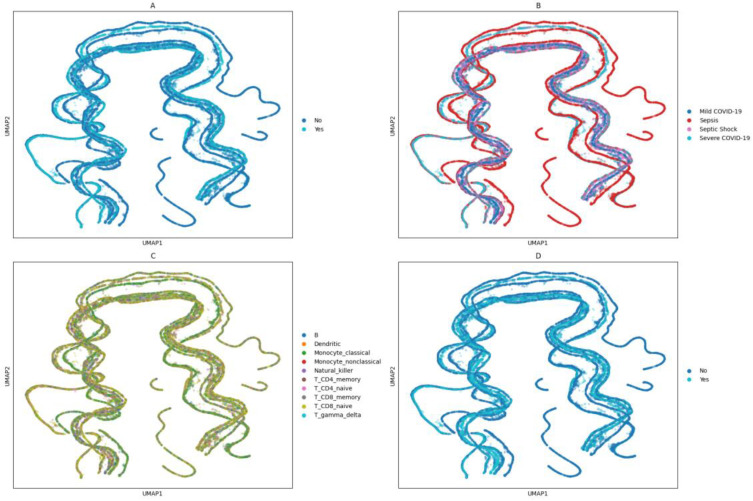
UMAP plots. A total of 19,002 cells from sepsis patients, 6268 cells from mild COVID-19 patients, 4276 cells from severe COVID-19 patients, and 8186 cells from septic shock patients expressing twenty immunologically associated gene transcripts (i.e., *FOSB*, *IGLC3*, *S100A9*, *S100A10*, *CD56*, *IFITM3*, *CD52*, *IL32*, *CD74*, *S100A12*, *HLA-A*, *CD24*, *CD8*, *CD14*, *CD19*, *TCRɣδ*, *CD3*, *TLR2*, *CD16*, and *CD4*) were mapped and shown in a two-dimensional plot. We separated different clinical information to elicit differentiated behavior in cells expressing transcripts, as seen in (**A**), which shows the cells under different medical conditions. (**B**) shows the cell types separated. (**C**) indicates whether the cell originated from a mild or severe patient. (**D**) conveys whether the patient’s infection is of a viral (mild and severe COVID-19) or non-viral (sepsis and septic shock) nature.

**Table 1 ijms-25-05920-t001:** Patient population characteristics.

	Sepsis	Mild COVID-19 *	Septic Shock	Severe COVID-19
Age (mean)	57.25	76.33	61.25	65.25
Weight (kg)	86.25	76.27	100	82.35
Sex at birth [m (*n*; %) f (*n*, %)]	m (3, 18.75%) f (1, 6.25%)	m (2, 12.5%) f (1, 6.25%)	m (3, 18.75%) f (1, 6.25%)	m (2, 12.5%) f (2, 12.5%)
PaO_2_/FiO_2_ (worst)	15	29	15	17
Temperature °C (highest)	37.88	37.47	39.7	37.3
CRRT (%)	25	0	25	50
Lactate (nmol/L)	2.3	2.77	2.86	2.43
Creatinine (µmol/L)	152.5	95.67	92.5	182
Bilirubin (mg/dL)	21	17	12	18
HB (g/dL)	10.33	11.67	7.1	12.35
APTT (seconds)	42.73	31.67	30.2	54.5
CRP (mg/L)	229.7	96.8	341.89	74.35
Neutrophils (10^9^/L)	25.43	5.17	14.9	10.85
Lymphocytes (10^9^/L)	0.38	0.77	0.8	0.98
APACHE (mean)	37	9	41	21
SOFA (mean)	9.75	1.3	12.75	7
Hospital length of stay (days)	124	31	149	17
Survivors (%)	75	66	75	75
ICU length of stay (days)	24.25	0	38.5	24.5
Bacterial pathogens identified	*K. oxytoca*; *S. aureus*	NA	*S. aureus* *S. epidermidis*	*S. epidermidis* *E. coli* *K. pneumoniae*

* Clinical data from one patient in the sepsis group was missing from the records of our clinical cohort. PBMC samples from this patient was normally included for performing ScRNA-seq.

**Table 2 ijms-25-05920-t002:** Different classification algorithms performance.

	Accuracy	Precision	Recall	Specificity
Support Vector Machine	0.81	0.62	0.47	0.82
Random Forest	0.82	0.69	0.49	0.83
XGBoost	0.88	0.73	0.71	0.91
Logistic Regression	0.69	0.63	0.15	0.95
Gradient Boosting	0.77	0.71	0.50	0.90
K-Nearest Neighbors	0.76	0.64	0.57	0.84

## Data Availability

The codes that perform the single-cell analysis and the XGBoost classification are publicly available at https://github.com/gustavsganzerla/immuno-scRNAseq (accessed on 24 May 2024). The gene expression matrices resulting from the single-cell RNA sequencing are available upon reasonable request to the corresponding author.

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
