# Peer review of "Identification of Marker Genes in Infectious Diseases from ScRNA-seq Data Using Interpretable Machine Learning"

_ijms, 2024, doi:10.3390/ijms25115920_

Round 1
Reviewer 1 Report
Comments and Suggestions for Authors
In their study, Martinez et al. introduce an approach aimed at effectively distinguishing between infectious conditions using machine learning techniques. The research is notable, and the technological steps undertaken are methodologically sound. However, I believe the paper exhibits several significant flaws.
The utilization of a single-cell approach enables researchers to detect alterations in specific cell populations and analyze these populations individually, considering their unique properties and gene expressions. However, it appears that the authors have adopted an approach more aligned with bulk transcriptomics rather than single-cell RNA sequencing. This resembles RNA deconvolution methodologies, where cell types are inferred based on specific gene expressions without proper characterization and quantification.
Notably, there is a lack of characterization of cell population clusters, which would facilitate the individual investigation of these clusters, as evident from the UMAP analysis. While their approach successfully identifies markers that differentiate between conditions, these markers could potentially have been discovered through bulk RNA sequencing.
This limitation diminishes the potential insights that could be gleaned from the data, resulting in deficient results and discussion sections that lack insights into the specific workings of individual cell populations. I believe the authors possess both the data and the tools necessary to approach this study differently, thereby significantly enhancing its scientific impact. Perhaps an approach that applies ML to individual cell populations can significantly increase the appeal of the paper.
In closing, a small note about the small size of the introduction section. It should be further expanded.
Comments on the Quality of English LanguageEnglish language is fine with a few poorly structured sentences which I am sure will be fixed with another pass.
Author Response
Reviewer 1
In their study, Martinez et al. introduce an approach aimed at effectively distinguishing between infectious conditions using machine learning techniques. The research is notable, and the technological steps undertaken are methodologically sound. However, I believe the paper exhibits several significant flaws.
Response: We appreciate being given the opportunity to enhance our study. We believe that after a meticulous peer review, our paper has improved a lot.
The utilization of a single-cell approach enables researchers to detect alterations in specific cell populations and analyze these populations individually, considering their unique properties and gene expressions. However, it appears that the authors have adopted an approach more aligned with bulk transcriptomics rather than single-cell RNA sequencing. This resembles RNA deconvolution methodologies, where cell types are inferred based on specific gene expressions without proper characterization and quantification. Notably, there is a lack of characterization of cell population clusters, which would facilitate the individual investigation of these clusters, as evident from the UMAP analysis. While their approach successfully identifies markers that differentiate between conditions, these markers could potentially have been discovered through bulk RNA sequencing. This limitation diminishes the potential insights that could be gleaned from the data, resulting in deficient results and discussion sections that lack insights into the specific workings of individual cell populations. I believe the authors possess both the data and the tools necessary to approach this study differently, thereby significantly enhancing its scientific impact. Perhaps an approach that applies ML to individual cell populations can significantly increase the appeal of the paper.
Response: We agree with the reviewer. We, initially, opted to use a Machine Learning (ML) based approach to deal with single cell data as our dataset would be composed of the expression profile of cells in different cell types. Initially, for the sake of simplicity, we opted not to include further information about cell populations expressing different genes as we obtained valuable clinical insights of transcripts that well characterized each disease model. After considering the valuable comments by the reviewer, we opted to include results of a ML step in which we use transcripts from single cells to classify the cell population they belong. This reflected in a new paragraph being included in our results (lines 285-297).
“To assess whether gene transcripts are good characterizers of different cell types, we performed an XGBoost multiclass classification where the inputs are scRNA-seq transcripts and the outputs are B, dendritic, monocyte classical, monocyte nonclassical, natural killer, TCD4 memory and naïve, TCD8 memory and naïve, and T gamma delta cells. Our model (Figure 3-A) could distinguish between all cell types with AUC scores ranging from 0.96 (highest, T gamma delta) to 0.75 (lowest, TCD8 memory). Moreover, we checked the individual contribution of transcripts to assign a label to a cell type of the classification models with AUC higher than 0.8. When looking at the gene transcripts that best characterized each cell type, we identified transcripts DEFA3 and FOSB to be the best characterizers of dendritic cells (Figure 3-C), monocyte classical cells (Figure 3-D), monocyte nonclassical cells (Figure 3-E), TCD4 memory cells (Figure 3G), TCD8 naïve cells (Figure 3-H), and T gamma delta cells (Figure 3-I). We also identified that expression of CD24 is a good characterizer of B cells (Figure 3-B)."
In closing, a small note about the small size of the introduction section. It should be further expanded.
Response: We appreciate the valid concern. We have modified our introduction an added more details on how AI can assist in scRNA-seq data analysis (lines 76-91).
“We highlight the Single cell RNA-sequencing (scRNA-seq) technique, which allows the analysis of gene expression at a cellular level. This method allows the analysis of the gene expression profile of individual cells within a population, which might generate a snapshot of gene expression in each individual cell. Many areas of research and clinical practice can benefit from scRNA-seq, such as cancer research, drug discovery and development, and immunology, among others. One remarkable translational approach of scRNA-seq is the development of precision therapeutic strategies [26]. To benefit from the depth of data analysis provided by scRNAseq, data analytics techniques are needed [27]. The analysis of genomics data might be powered by Artificial Intelligence (AI) techniques, which hold the mathematical power to untangle the intricate relationship between datapoints in higher dimensional settings [28]. Examples of AI assisting medicine in general can be found in the identification of marker genes [7-8], biomarkers [9-10], vaccine candidates [11], and potential targets for therapeutics [12]. Moreover, AI has been a companion for scRNA-seq data analysis such as identification of tumor cells [29], and , AI generative models have also been employed in predicting the expression level of genes in a single cell [30].”
Reviewer 2 Report
Comments and Suggestions for Authors
Despite studying an important area, i.e. the immunobiology of both sepsis and COVID-19 infection in different severities of disease, and using advanced scRNASeq and machine learning techniques, the description of results and discussion in the manuscript lacks clarity and consistency. Clinical information of the 4 patients in each class is missing, and this might have important impact on the interpretation of the results, as sepsis is inherently heterogeneous, and a small sample size might limit external validity of the results. Also, the discussion on the top differentially expressed/class-identifying genes, lacks depth and specific interpretation in the context of sepsis or COVID-19. In my view, the manuscript requires major revisions to allow correct assessment of the data and the author's interpretation hereof. Also, the language mistakes result in a frustrating experience for the reader, this should be improved as well.
Detailed remarks:
- The introduction and/or discussion section lacks a reference to a general article discussing the role of scRNAseq in deciphering sepsis immunobiology, as well as a discussion of the advantages and potential caveats of using machine learning versus other (e.g. deep learning) approaches to the analysis of scRNAseq data.
- Can the authors state why they chose to test RF, XGBoost and SVM algorithms? Specifically, why was SVM tested despite a dataset with four (rather than 2) classes?
- The results section contains an error. Both subtitle 2.1 and 2.2 read "using scRNA-seq transcripts as an input to classify diagnostics"
- Starting the results section with a description of the patient characteristics, number of cells analyzed, subsets of cells present etc would increase clarity. Also, given the limited number of patients per group, it is relevant to know how heterogeneous they were (e.g. different time point of sampling in disease history, different etiologies of bacterial sepsis) and how "mild" and "severe" were defined. How clear is the separation between "COVID" and "sepsis": could patients with COVID-19 have had bacterial co/superinfection resulting in sepsis, or not? The septic patients with "single organ failure", were these all respiratory failure (in analogy with the COVID-19 patients with "single organ failure" most likely having respiratory failure)? What were the details (bacterial etiology + organ focus) of the bacterial sepsis patients?
- Results line 91 and below: it is very difficult to decipher from the manuscript text and the figure whether the named transcripts are upregulated or downregulated in the 4 diagnostic classes of patients.
- Results contain seemingly contradictory information:
*line 95 suggests that high levels of CD16 are characteristic of COVID-19 (compared to sepsis), however this difference is not very pronounced in the severe forms of COVID-19 and sepsis (see Figure 1F), respectively.
*line 97 suggests that high expression of TCRgammadelta was characteristic of sepsis, while the difference in TCR gamma delta expression is not very pronounced in the severe forms of COVID-19 and sepsis (see Figure 1F), respectively.
* line 124 (and 146) suggest CD16 has high expression in sepsis, while Figure 1F suggests the expression is lowest in the sepsis group, compared to septic shock and both COVID-19 groups.
* line 138 suggests TCRgamma delta is upregulated in severe COVID-19 patients compared to patients with bacterial sepsis, which contrasts with line 97.
_ The UMAP plots (Figure 4) have a "line" appearance, rather than the typical dot appearance. UMAP plot B does not display all colours as shown in the legend.
- The discussion lacks depth, nuance and clarity regarding the role of the identified transcripts in the disease processes of sepsis/septic shock and COVID-19 infection.
(a) "here we argue the elevated expression of CD14 in septic shock is a result of CD14+ cells binding with LPS". LPS is not involved in all cases of sepsis (e.g. gram-positive sepsis). Also, the next lines (243 and below) suggest the severity of disease is explained by a difference in microbial load, while in many instances, the severity of the dysregulated host response is not solely, let alone mainly, determined by the microbial load.
(b) "we hypothesize the lower FOSB expression might suppress the expression of genes that ... play a detrimental role in controlling an infection" Do the authors mean FOSB low -> genes (which?) expression low -> less negative effect on controlling the infection, i.e. better infection control. This is nonsensical.
(c) "in a viral scenario, gamma delta + T cells were found in rise": this refers to an article on dengue infection, with a pathogenesis that is supposedly very different from severe COVID-19.
- "Informed consent: not applicable?" Patient or proxy consent for blood sampling (for PBMCs) and collection of basic clinical data should have been sought.
Comments on the Quality of English LanguageExtensive language editing should be done, as many parts of the manuscript are difficult to understand.
e.g. abstract line 17 "the immune system might calibrate" : what is meant here?
abstract line 19-20 "...immune response can reflect in differentiated immune cell populations, clinical indicators and ..." : differentiated or differential expansion of? clinical indicators is very vague.
Introduction line 35: "this reality became more pronounced also including viruses when the world ..." please rephrase.
Introduction line 39: " and has been proposed as a trigger" : what has been proposed?
introduction line 53-55: "despite its translational approach" : despite?
introduction line 61-62: "will leave a footprint in the transcriptomic level of ... cells" : level?
results - 2.1 line 71: "to classify diagnostics"
results 2.1 line 96: this inverse pattern
results 2.1 line 96: inversely
results 2.2 line 144: classify severity and mild disease: severe?
results figure 2: the legend contains segments of text that are exactly the same as in the manuscript body.
results 2.4: "the cell-expression" "different diagnostics"
results figure 4 line 213: "the medical condition of the cell"
discussion line 220-221: "we bring attention to the interpretation ... might results in potential marker genes ...whose immune mechanism can better explain the immune landscape of the body ... " please rephrase: difficult to understand, vague wording, incorrect grammar.
discussion line 240: "or the cells they infect producing signals that belong to DAMPs"
discussion line 243: "as in general septic shock patients demonstrate microbial load"
discussion line 259: " the FOSB expression is inversely related to prognosis": to good or bad prognosis? associated seems more correct than related.
discussion line 266: remove (Meijer et al, 2012)
discussion line 285: outside of the universe?
discussion line 289: is appreciated? is needed?
discussion line 293: severe symptoms of : presentations of?
methods line 334: that are common of: that were shared between?
methods line 337: "diagnostics"
methods line 367: " the explained was applied in the train data" ??
methods line 368: typo extent rather than extend
" we measured the extend of which the features was consider to assign a label to each class": rephrase
Author Response
Reviewer 2
Despite studying an important area, i.e. the immunobiology of both sepsis and COVID-19 infection in different severities of disease, and using advanced scRNASeq and machine learning techniques, the description of results and discussion in the manuscript lacks clarity and consistency. Clinical information of the 4 patients in each class is missing, and this might have important impact on the interpretation of the results, as sepsis is inherently heterogeneous, and a small sample size might limit external validity of the results. Also, the discussion on the top differentially expressed/class-identifying genes, lacks depth and specific interpretation in the context of sepsis or COVID-19. In my view, the manuscript requires major revisions to allow correct assessment of the data and the author's interpretation hereof. Also, the language mistakes result in a frustrating experience for the reader, this should be improved as well.
Response: We appreciate the detailed review. We believe that after having incorporated all the comments in our manuscript, it substantially improved.
Detailed remarks:
- The introduction and/or discussion section lacks a reference to a general article discussing the role of scRNAseq in deciphering sepsis immunobiology, as well as a discussion of the advantages and potential caveats of using machine learning versus other (e.g. deep learning) approaches to the analysis of scRNAseq data.
Response: Thank you for the valid concern, we added a new paragraph in our introduction providing details on how single cell RNA sequencing might benefit from AI-based data analysis (lines 76-91).
“We highlight the Single cell RNA-sequencing (scRNA-seq) technique, which allows the analysis of gene expression at a cellular level.This method allows the analysis of the gene expression profile of individual cells within a population, which might generate a snapshot of gene expression in each individual cell. Many areas of research and clinical practice can benefit from scRNA-seq, such as cancer research, drug discovery and development, immunology, among others. One remarkable translational approach of scRNA-seq is the development of precision therapeutic strategies [26]. In order to benefit from the depth of data analysis provided by scRNAseq, data analytics techniques are needed [27]. The analysis of genomics data might be powered by Artificial Intelligence (AI) techniques, which hold the mathematical power to untangle the intricate relationship between datapoints in higher dimensional settings [28]. Examples of AI assisting medicine in general can be found in the identification of marker genes [7-8], biomarkers [9-10], vaccine candidates [11], and potential target for therapeutics [12]. Moreover, AI has been a companion for scRNA-seq data analysis such as identification of tumor cells [29]. Moreover, AI generative models have also been employed in predicting the expression level of genes in a single cell [30].”
- Can the authors state why they chose to test RF, XGBoost and SVM algorithms? Specifically, why was SVM tested despite a dataset with four (rather than 2) classes?
Response: We appreciate having the opportunity to clarify. For having a more representative suite of algorithms, we added Logistic Regression, Gradient Boosting, and K-Nearest Neighbors. We used a one vs. all approach to perform the multiclass classification. We have added such details in the text (lines 146-149).
“We created a function of 20 transcripts to classify diagnostics of patients (i.e., mild-sepsis, mild-COVID-19, septic shock, and severe COVID-19) in a one vs. all ap-proach. Upon testing the classification capacity of RF, SVM, XGBoost, LR, GB, and KNN, we report the best classification metrics were achieved by XGBoost (Table 2).”
- The results section contains an error. Both subtitle 2.1 and 2.2 read "using scRNA-seq transcripts as an input to classify diagnostics"
Response: We appreciate the reviewer noticing this problem in our text. We have fixed it.
- Starting the results section with a description of the patient characteristics, number of cells analyzed, subsets of cells present etc would increase clarity. Also, given the limited number of patients per group, it is relevant to know how heterogeneous they were (e.g. different time point of sampling in disease history, different etiologies of bacterial sepsis) and how "mild" and "severe" were defined. How clear is the separation between "COVID" and "sepsis": could patients with COVID-19 have had bacterial co/superinfection resulting in sepsis, or not? The septic patients with "single organ failure", were these all respiratory failure (in analogy with the COVID-19 patients with "single organ failure" most likely having respiratory failure)? What were the details (bacterial etiology + organ focus) of the bacterial sepsis patients?
Response: We agree with the reviewer, by adding a clinical table, we are providing more context to the cohort of patients we have. This resulted in the inclusion of a new section in our results (lines 126-141).
“2.1. Clinical characteristics of the cohort
We acquired data from 16 patients, 4 of whom had sepsis, 4 had moderate COVID-19, 4 had septic shock, and 4 had severe COVID-19 (Table 1). We found the average age of the patients was 65 years of age, The male-to-female ratio in this cohort is roughly two to one. Recordings showed temperatures higher than 37°C across the board with the septic shock population showing the highest mean (39.7°C). Patients with mild COVID-19 had the highest P/F ration while those with sepsis and septic shock had the lowest mean. Neutrophil numbers were found to be the lowest in moderate COVID-19 and the greatest in sepsis. The sepsis group had the lowest mean lymphocyte count. The lowest Sequential Organ Failure Assessment (SOFA) score was observed in individuals with mild COVID-19 and the highest in septic shock. It is worth noting that severe COVID-19 patients had a SOFA score that was even lower than those suffering from sepsis.
Table 1 – Patient population characteristics
|
Sepsis |
Mild COVID-19* |
Septic shock |
Severe COVID-19 |
Age (mean) |
57.25 |
76.33 |
61.25 |
65.25 |
Weight (kg) |
86.25 |
76.27 |
100 |
82.35 |
Sex at birth [m (n; %) f (n, %)] |
m (3, 18.75%) f (1, 6.25%) |
m (2, 12.5%) f (1, 6.25%) |
m (3, 18.75%) f (1, 6.25%) |
m (2, 12.5%) f (2, 12.5%) |
PaO2/FiO2 (worst) |
15 |
29 |
15 |
17 |
Temperature °C (highest) |
37.88 |
37.47 |
39.7 |
37.3 |
CRRT (%) |
25 |
0 |
25 |
50 |
Lactate (nmol/L) |
2.3 |
2.77 |
2.86 |
2.43 |
Creatinine (µmol/L) |
152.5 |
95.67 |
92.5 |
182 |
Bilirubin (mg/dL) |
21 |
17 |
12 |
18 |
HB (g/dL) |
10.33 |
11.67 |
7.1 |
12.35 |
APTT (seconds) |
42.73 |
31.67 |
30.2 |
54.5 |
CRP (mg/L) |
229.7 |
96.8 |
341.89 |
74.35 |
Neutrophils (109/L) |
25.43 |
5.17 |
14.9 |
10.85 |
Lymphocytes (109/L) |
0.38 |
0.77 |
0.8 |
0.98 |
APACHE (mean) |
37 |
9 |
41 |
21 |
SOFA (mean) |
9.75 |
1.3 |
12.75 |
7 |
Hospital length of stay (days) |
124 |
31 |
149 |
17 |
Survivors (%) |
75 |
66 |
75 |
75 |
ICU Length of stay (days) |
24.25 |
0 |
38.5 |
24.5 |
Bacterial pathogens identified |
K. oxytoca; S. aureus |
NA |
S. aureus S. epidermidis |
S. epidermidis E. coli |
”
- Results line 91 and below: it is very difficult to decipher from the manuscript text and the figure whether the named transcripts are upregulated or downregulated in the 4 diagnostic classes of patients.
Response: We appreciate the important remark of the reviewer. We have modified Figure 1-F to better display the levels of each transcripts that characterized each class of patients.
- Results contain seemingly contradictory information:
*line 95 suggests that high levels of CD16 are characteristic of COVID-19 (compared to sepsis), however this difference is not very pronounced in the severe forms of COVID-19 and sepsis (see Figure 1F), respectively.
*line 97 suggests that high expression of TCRgammadelta was characteristic of sepsis, while the difference in TCR gamma delta expression is not very pronounced in the severe forms of COVID-19 and sepsis (see Figure 1F), respectively.
* line 124 (and 146) suggest CD16 has high expression in sepsis, while Figure 1F suggests the expression is lowest in the sepsis group, compared to septic shock and both COVID-19 groups.
* line 138 suggests TCRgamma delta is upregulated in severe COVID-19 patients compared to patients with bacterial sepsis, which contrasts with line 97.
Response: We appreciate the important points raised by the reviewer. In fact, we initially failed to properly document the levels of transcription that best characterized each disease model. Thus, we opted to redesign Figure 1-F to make it easier to interpret. We also reshaped the paragraph in which we report these results (lines 167-190).
“First, we identified that the higher expression of CD3 is a characteristic of the severe forms of sepsis and COVID-19; this same pattern was observed for the gene CD14. Next we report that the lowest levels of CD16 were found in cells from sepsis patients while the highest levels were found in mild COVID-19 patients. The gene FOSB was found to be expressed in higher amounts in the mild forms of sepsis and COVID-19. Finally, a similar pattern was found for the expression of genes S100A12 and T-cell receptors É£δ (TCRÉ£δ), which were highly expressed in the bacterial forms of infection and lowly expressed in the viral infections.”
_ The UMAP plots (Figure 4) have a "line" appearance, rather than the typical dot appearance. UMAP plot B does not display all colours as shown in the legend.
Response: We appreciate the valuable feedback on how to improve our UMAP plots. We have provided clearer images. We believe that the appropriate visualization was hindered by the lower quality of figures we initially proposed.
- The discussion lacks depth, nuance and clarity regarding the role of the identified transcripts in the disease processes of sepsis/septic shock and COVID-19 infection.
Response: We appreciate the valid concern of the reviewer. We have incorporated the comments below in order to make our discussion deeper.
(a) "here we argue the elevated expression of CD14 in septic shock is a result of CD14+ cells binding with LPS". LPS is not involved in all cases of sepsis (e.g. gram-positive sepsis). Also, the next lines (243 and below) suggest the severity of disease is explained by a difference in microbial load, while in many instances, the severity of the dysregulated host response is not solely, let alone mainly, determined by the microbial load.
Response: We appreciate the feedback. We have reshaped this paragraph in order to make the language more conservative (lines 547-553).
“We suggest that the increased presence of CD14 could potentially indicate severity in certain conditions. First, in septic shock, it is believed that CD14-positive cells binding with LPS (lipopolysaccharides) play a role, particularly as septic shock patients often show high microbial levels. Secondly, in severe cases of COVID-19, it is proposed that the high levels of viral RNA in the bloodstream necessitate more CD14-positive cells to identify these viral molecular patterns, as indicated in a study by [16]”
(b) "we hypothesize the lower FOSB expression might suppress the expression of genes that ... play a detrimental role in controlling an infection" Do the authors mean FOSB low -> genes (which?) expression low -> less negative effect on controlling the infection, i.e. better infection control. This is nonsensical.
Response: We appreciate the constructive critique. In fact, the wording we used was not appropriate. We have rewritten this passage to reflect the lower expression of AP-1 might be an indicator of a dysregulated immune response (lines 566-569).
“According to our data, the FOSB expression is inversely associated with prognosis; thus it might be the case that lower FOSB expression is preventing potential pathways that lead to a repressed dysregulated immune response.”
(c) "in a viral scenario, gamma delta + T cells were found in rise": this refers to an article on dengue infection, with a pathogenesis that is supposedly very different from severe COVID-19.
Response: We appreciate the feedback of the reviewer. We have used a COVID-19 specific reference for this passage.
- "Informed consent: not applicable?" Patient or proxy consent for blood sampling (for PBMCs) and collection of basic clinical data should have been sought.
Response: We apologize that. We in fact had obtained consent. We initially failed to include that outside the submission portal. The reviewed manuscript contains it (lines 829-831)
“As this is a project involved patients lacking capacity, next-of-kin assent was obtained with a witness present prior to enrollment in the study. Written consent was reaffirmed for patients lacking capacity once they regained capacity.”
Comments on the Quality of English Language
Extensive language editing should be done, as many parts of the manuscript are difficult to understand.
Response: We appreciate the reviewer pointing specific details on how we can improve the writing of our work. We have carefully addressed each of the points.
e.g. abstract line 17 "the immune system might calibrate" : what is meant here?
Response: We appreciate the concern, we replaced “calibrate” with “undergo regulation” to better reflect our writing goals with the passage.
abstract line 19-20 "...immune response can reflect in differentiated immune cell populations, clinical indicators and ..." : differentiated or differential expansion of? clinical indicators is very vague.
Response: We appreciate the concern. We removed the vague term.
Introduction line 35: "this reality became more pronounced also including viruses when the world ..." please rephrase.
Response: We agree with the reviewer, this sentence does not read well. We modified it (lines 35-38).
“Infectious diseases remain a substantial challenge for global health, a reality that was underscored by the emergence of the most devastating pandemic in over a century. This pandemic was caused by the infectious agent SARS-CoV-2, which brought viruses into sharper focus as a significant threat to public health worldwide”
Introduction line 39: " and has been proposed as a trigger" : what has been proposed?
Response: We appreciate the opportunity to clarify. We wanted to point that cytokine storm might result in multi organ failure. We modified the passage (lines 40-41).
“Moreover, cytokine storms may contribute to multi-organ failure.”
introduction line 53-55: "despite its translational approach" : despite?
Response: We appreciate the constructive feedback, we modified the sentence by removing the despite clause as it did not add to the passage and reshaped the paragraph that initially contained it (lines 76-91).
“We highlight the Single cell RNA-sequencing (scRNA-seq) technique, which allows the analysis of gene expression at a cellular level. This method allows the analysis of the gene expression profile of individual cells within a population, which might generate a snapshot of gene expression in each individual cell. Many areas of research and clinical practice can benefit from scRNA-seq, such as cancer research, drug discovery and de-velopment, and immunology, among others. One remarkable translational approach of scRNA-seq is the development of precision therapeutic strategies [26]. To benefit from the depth of data analysis provided by scRNAseq, data analytics techniques are needed [27]. The analysis of genomics data might be powered by Artificial Intelligence (AI) tech-niques, which hold the mathematical power to untangle the intricate relationship between datapoints in higher dimensional settings [28]. Examples of AI assisting medicine in general can be found in the identification of marker genes [7-8], biomarkers [9-10], vaccine candidates [11], and potential targets for therapeutics [12]. Moreover, AI has been a companion for scRNA-seq data analysis such as identification of tumor cells [29], and , AI generative models have also been employed in predicting the expression level of genes in a single cell [30].”
introduction line 61-62: "will leave a footprint in the transcriptomic level of ... cells" : level?
Response: We agree with the reviewer on the passage that is hard to read. We have modified it (lines 92-94).
“Here we hypothesize that the immune dysregulation caused by moderate and severe instances of viral and bacterial induced sepsis will reflect in different transcriptomics of different populations of immune cells”
results - 2.1 line 71: "to classify diagnostics"
Response: We appreciate the reviewer in noticing this issue with our writing. We have modified it (line 146).
“We created a function of 20 transcripts to classify diagnostics of patients (i.e., mild-sepsis, mild-COVID-19, septic shock, and severe COVID-19)”
results 2.1 line 96: this inverse pattern
Response: Thank you for pointing this out, we modified it in our text (lines 167-190).
“First, we identified that the higher expression of CD3 is a characteristic of the severe forms of sepsis and COVID-19; this same pattern was observed for the gene CD14. Next, we report that the lowest levels of CD16 were found in cells from sepsis patients while the highest levels were found in mild COVID-19 patients. The gene FOSB was found to be expressed in higher amounts in the mild forms of sepsis and COVID-19. Finally, a similar pattern was found for the expression of genes S100A12 and T-cell receptors É£δ (TCRÉ£δ), which were highly expressed in the bacterial forms of infection and lowly expressed in the viral infections.”
results 2.1 line 96: inversely
Response: Thank you for the feedback. As the reviewer kindly suggested for us to modify the previous sentence, this caused in the consequent modification of this as well.
results 2.2 line 144: classify severity and mild disease: severe?
Response: We appreciate the constructive critique, this has been modified (lines 160-162).
“We report that the input transcripts successfully classified the mild-sepsis, mild-COVID-19, septic shock, and severe COVID-19 patients in a one vs. all approach by indicating AUC scores of 0.93, 0.91, 0.89, and 0.96, respectively (Figure 1-A).”
results figure 2: the legend contains segments of text that are exactly the same as in the manuscript body.
Response: We appreciate the valid critique; we have rewritten the caption of the figure (lines 276-283).
“Figure 2 – The study compared hub gene expression in mild (Figure 2-A and 2-B) and severe cas-es (Figure 2-C and 2-D) of viral and bacterial-induced sepsis. We found that certain genes, like CD16:3G8|FCGR3A|AHS0053|PABO, were more expressed in mild COVID-19 patients and sepsis, while CD3:SK7|CD3E|AHS0033|PABO showed higher levels in septic shock and severe COVID-19 cases. S100A12 had low expression in mild COVID-19 and none in severe cases, mak-ing it a potential target in viral-induced diseases. TCR-GAMMA_DELTA: B1|TRD_TRG|AHS0015|PABO was upregulated in severe COVID-19, particularly in dendritic cells, suggesting its role as a classifier in critically ill COVID-19 patients.”
results 2.4: "the cell-expression" "different diagnostics"
Response: We agree with the reviewer, we could have provided an easier to read subtitle. It has been amended in the manuscript (line 407).
“Summary of cellular expression patterns for various diagnostics in a two-dimensional format.”
results figure 4 line 213: "the medical condition of the cell"
Response: Thank you for noticing this issue with our writing, we have amended it (lines 481-482).
“Figure 5-A shows the cells under different medical conditions.”
discussion line 220-221: "we bring attention to the interpretation ... might results in potential marker genes ...whose immune mechanism can better explain the immune landscape of the body ... " please rephrase: difficult to understand, vague wording, incorrect grammar.
Response: Thank you for noticing this issue with our writing, we have modified it (lines 488-490).
“Here we highlight the importance of interpreting these transcript signals, as they may lead to the identification of potential marker genes associated with severity.”
discussion line 240: "or the cells they infect producing signals that belong to DAMPs"
Response: We appreciate the feedback. We revisited this passage (lines 505-547).
“Upon viral infection, CD14+ cells also express other pattern recognition receptors that identify specific viral signals, such as their unique genetic material configuration. These signals are part of the Pathogen-Associated Molecular Patterns (PAMPs), or they may arise from infected cells producing signals categorized as Damage-Associated Molecular Patterns (DAMPs) [14].”
discussion line 243: "as in general septic shock patients demonstrate microbial load"
Response: We appreciate the reviewer pointing this out. As a result of the modifications suggested by the reviewer in the query regarding CD14, this whole passage has been modified (lines 547-553).
“We suggest that the increased presence of CD14 could potentially indicate severity in certain conditions. First, in septic shock, it is believed that CD14-positive cells binding with LPS (lipopolysaccharides) play a role, particularly as septic shock patients often show high microbial levels. Secondly, in severe cases of COVID-19, it is proposed that the high levels of viral RNA in the bloodstream necessitate more CD14-positive cells to identify these viral molecular patterns, as indicated in a study by [16].”
discussion line 259: " the FOSB expression is inversely related to prognosis": to good or bad prognosis? associated seems more correct than related.
Response: We agree with the reviewer, the term associated makes more sense. We have modified it in the manuscript (line 556).
“According to our data, the FOSB expression is inversely associated to prognosis, thus […]”
discussion line 266: remove (Meijer et al, 2012)
Response: Thank you for pointing this misuse of reference that was overlooked during our final drafting. We promptly removed it.
discussion line 285: outside of the universe?
Response: We are in line with the reviewer as the use of the term “the universe” might be too vague here. We modified this passage (lines 591-593).
“we argue that the complexity of immune responses could be reflected in transcriptional changes in genes beyond the initial scope inquired by our ScRNA-seq analysis, providing a more comprehensive perspective on the expression levels of the analyzed patients.”
discussion line 289: is appreciated? is needed?
Response: In fact, we agree with the replacement of the term. Please find it modified in lines 596-597.
“an in-depth review of their role in the context of infection is needed.”
discussion line 293: severe symptoms of : presentations of?
Response: Thank you for this kind suggestion, we modified the terminology (lines 676-679).
“The synergy between these methodologies has the potential to shed light on the processes that are responsible for immune responses by identifying transcriptional alterations in genes of interest during moderate and severe presentations of infectious diseases.”
methods line 334: that are common of: that were shared between?
Response: We appreciate the suggestion, we modified as pointed by the reviewer (lines 716-719).
“Out of an immunological panel with 400 immune-related transcripts, we extracted the 20 gene transcripts (i.e., FOSB, IGLC3, S100A9, S100A10, CD56, IFITM3, CD52, IL32, CD74, S100A12, HLA-A, CD24, CD8, CD14, CD19, TCRÉ£δ, CD3, TLR2, CD16, and CD4) that are shared between all the 16 patients.”
methods line 337: "diagnostics"
Response: We have promptly modified the term (lines 720-722).
“These matrices were treated with in house Python (version 3.9) scripts to merging the matrices, finding and sub-setting the matrices as per the transcripts in common, mapping the cell index into 10 different cell types, adding a column related to the parameters, and normalizing the data”
methods line 367: " the explained was applied in the train data" ??
Response: We appreciate the reviewer noticing this typo in our writing, we have corrected it (line 766).
“The explanation process was applied in the train data”
methods line 368: typo extent rather than extend
Response: Thank you for noticing this issue in our writing, it has been amended (lines 767).
“We measured the extent of which the features”
" we measured the extend of which the features was consider to assign a label to each class": rephrase
Response: In fact, the initial writing style of the passage was confusing, we rephrased it (lines 766-767).
“We assessed the extent to which the features (i.e., 20 immunological transcripts) were used for assigning a label to each class”
Round 2
Reviewer 1 Report
Comments and Suggestions for Authors
Dear authors, I appreciate your efforts in improving the manuscript and the significant changes in the way you present your work.
However, I still believe that your work lacks methodological soundness, not in the machine learning part but rather in how you approach single-cell data. You say you characterized cell population but how did you achieve that? There has to be a section in which you describe cluster characterization. This doesn't exist either in your GitHub code or your methodology section. I would strongly suggest that you explore more papers that use scRNA-seq data and try to formulate a better approach.
Your hypothesis is interesting and the analyses presented here have merit but neither strongly support the conclusions nor present all data/methods effectively. Improving upon the language of the manuscript has helped a lot but cannot account for a lackluster methodology.
I know that this might seem harsh but as I've previously said you have shown that you have valuable data and the technical knowledge to perform ML analyses. You just need to explore how to handle sc-RNA seq in a more efficient manner. Try to look into pipelines like Seurat or Harmony and how those approach such data. These will help to better understand and explain your data before applying ML so that your conclusions have a stronger impact.
Author Response
Response: Dear reviewer, we appreciate your efforts in suggesting improvements in our work. We believe that after review, our paper has substantially improved. We also understand that traditional single cell RNA-seq data analysis approaches are valid ways to retrieve meaningful clinical interpretation on cells undergoing different stressors. However, this was never our goal with this submission. We would like to clarify that our objective for this paper has always been to propose an Artificial Intelligence based method for assisting in the analysis of single cell RNA-seq data, which can further be leveraged to identify marker genes of severity in sepsis, septic shock, mild COVID-19, and severe COVID-19. We believe that by treating each cell as a row and their associated transcript a column, we can entertain a classification system in which AI be used to extract meaningful information. We understand that we might have failed to convey this message in a clearer manner to readers, so we opted to reshape the last paragraph of our introduction in which we present our goals (lines 82-88).
“In our work, we hypothesize that AI has the capacity to untangle patterns and capture signals of individual cells expressing different transcripts when stressed by moderate and severe manifestations of viral and bacterial induced sepsis. Moreover, by mapping the decision patterns of AI tools, we can generate valuable insights into the intricate immunological responses deployed by the body in its battle against infectious pathogens, highlighting essential marker genes for advancing therapeutic interventions.”
Round 3
Reviewer 1 Report
Comments and Suggestions for Authors
Dear authors, I believe that we still hold different opinions. You claim that your ML approach can characterize cell types but don't really explain how you figure out that population A is for example B cells. How does your model know to attach these "tags"? Did you provide some different data, not described here, in which you first tell the model what the transcriptome of B cells should look like?
You say that "In order to assess whether gene transcripts are a good characterizer of different cell types, we performed an XGBoost multiclass classification". Of course gene transcripts can characterize cell types. That's the whole point behind scRNA-seq cell type characterization and its various approaches. If you want to convey your message which states "In our work, we hypothesize that AI has the capacity to untangle patterns and capture signals of individual cells expressing different transcripts when stressed by moderate and severe manifestations of viral and bacterial induced sepsis. this should be something like '' In order to assess whether gene transcripts can characterize efficiently the viral infection -induced changes inside specific cell types".
Let's break down the methodology for achieving something like this:
1) Obtain and sequence samples
2) Characterize each cell type using known markers.
3) Explore the changes in cell types during infection
4) Propose ML fetures which characterize the infected cells
I still think step 2 is either missing or not well-explained.
Comments on the Quality of English Language
In addition you new edits have introduced some new typographical mistakes like "F/P ration" which I believe is ratio.
Author Response
Dear reviewer,
We appreciate your effort to try to clarify some methodological aspects of our work.
We understand our methodological approach might be better explained to readers, for that, we have made another attempt to make our methodology clearer. We reshaped the paragraph in our methodology in which we display this aspect of the study design (lines 571-581)
“We obtained single cell expression matrices from 16 patients. Each row of the matrix represents a single cell. The matrices are composed of 400 columns representing a panel with 400 immune-related transcripts from the we extracted the transcripts that are shared between all patients (variables x). From a classification point of view, each cell needs to have a variable y representing its label which is inherent to the data. We employ two distinct classification routines, using the complex transcriptional signal, represented by the transcripts x to classify: i) a y cell type, ranging from B cell, dendritic, monocyte classical, monocyte nonclassical, natural killer, TCD4 memory, TCD8 naïve, and T gamma delta; and ii) a y medical condition the cells are undergoing, i.e., sepsis, mild COVID-19, septic shock, and severe COVID-19. Both classification steps I and ii are denoted in Equations 1 and 2, respectively.
(Equation 1)
(Equation 2)”
In our dataset, each cell is characterized by a specific cell type label obtained directly from the processing of single-cell RNA sequencing (scRNA-seq) data. These labels serve as the target variable, denoted as 'y', in our supervised machine learning approach.
Supervised machine learning, often employed in classification tasks, requires known labels to evaluate the performance of the model. In our case, we utilize a data matrix where each row corresponds to a cell, and each column represents a transcript. The final column ('y') contains the cell type labels, representing a multiclass classification problem (e.g., B cell, NK cell, monocytes, etc.).
Our objective is to derive a function, denoted as 'f(x) = y', where 'x' represents the combination and magnitudes of transcript expression levels. Through supervised learning, we aim to train our model to accurately predict cell type labels based on transcriptomic profiles.
During model evaluation, we compare the predicted labels generated by our model with the actual labels to assess its performance. This comparison allows us to assess the model's ability to correctly classify cells into their respective categories.
We also appreciate the feedback regarding the typo of 'ration'. We have fixed this in lines 129-130.
“Patients with mild COVID-19 had the highest P/F ratio while those with sepsis and septic shock had the lowest mean.”
Please let us know it the current version of the manuscript addressed your concerns.
Regards,
Gustavo Sganzerla Martine (on behalf of the authors)
Round 4
Reviewer 1 Report
Comments and Suggestions for Authors
Dear authors,
am I to understand that cell type labels were provided by the sequencing platform? You state "In our dataset, each cell is characterized by a specific cell type label obtained directly from the processing of single-cell RNA sequencing (scRNA-seq) data. These labels serve as the target variable, denoted as 'y', in our supervised machine learning approach." This is still not clear. This is what i have been asking from the beginning. What is the processing which provides the labels "B cell, dendritic, monocyte classical, monocyte nonclassical, natural killer, TCD4 memory, TCD8 naïve, and T gamma delta; " for each of your cells on which you train your model ?
Author Response
Dear reviewer,
we appreciate your effort in making key suggestions for us to deliver our message in the clearest possible way. We confirm the cell type label comes directly from the sequencing platform. In order to consolidate this, we added a passage in our methodology (lines 581-584).
"Both labels (cell type and medical condition) are already present in the data, where the cell types are obtained from the BD Rhapsody Single-Cell Analysis System platform and the medical condition matches the patient the sample came from."